# Micrometer-thick and porous nanocomposite coating for electrochemical sensors with exceptional antifouling and electroconducting properties

Jeong-Chan Lee[1,2,11], Su Yeong Kim[2,11], Jayeon Song[3,4,5,11], Hyowon Jang[3], Min Kim[2], Hanul Kim [6], Siyoung Q. Choi [6], Sunjoo Kim[7], Pawan Jolly [1], Taejoon Kang [3,8] ✉, Steve Park [2] ✉ & Donald E. Ingber [1,9,10] ✉

Development of coating technologies for electrochemical sensors that consistently exhibit antifouling activities in diverse and complex biological environments over extended time is vital for effective medical devices and diagnostics. Here, we describe a micrometer-thick, porous nanocomposite coating with both antifouling and electroconducting properties that enhances the sensitivity of electrochemical sensors. Nozzle printing of oil-in-water emulsion is used to create a 1 micrometer thick coating composed of cross-linked albumin with interconnected pores and gold nanowires. The layer resists biofouling and maintains rapid electron transfer kinetics for over one month when exposed directly to complex biological fluids, including serum and nasopharyngeal secretions. Compared to a thinner (nanometer thick) antifouling coating made with drop casting or a spin coating of the same thickness, the thick porous nanocomposite sensor exhibits sensitivities that are enhanced by 3.75- to 17-fold when three different target biomolecules are tested. As a result, emulsion-coated, multiplexed electrochemical sensors can carry out simultaneous detection of severe acute respiratory syndrome coronavirus 2 (SARS-CoV-2) nucleic acid, antigen, and host antibody in clinical specimens with high sensitivity and specificity. This thick porous emulsion coating technology holds promise in addressing hurdles currently restricting the application of electrochemical sensors for point-of-care diagnostics, implantable devices, and other healthcare monitoring systems.

Bioelectronic devices, such as electrochemical sensors used for medical diagnostics, have witnessed remarkable growth, finding diverse applications in healthcare, energy, and environmental monitoring[1–3]. However, biofouling, the unwanted accumulation of biological materials on electrodes, presents a key challenge for the commercial development of bioelectronic devices because it leads to performance and reliability issues due to inaccurate electrode functioning in biofuel cells, medical implants, and biosensors[4,5]. Biofouling also can trigger immune responses and infections when devices contact living tissues[6,7]. Therefore, practical strategies to mitigate biofouling are crucial for improving electronic device performance if they are to be successfully applied to solve critical medical diagnostic and sensing challenges.

---

Several antifouling technologies have been developed to combat biofouling[6]. These include physical barriers, such as membranes or filters, to prevent fouling agents from reaching the device surface[8]. Other approaches include using tailored microtopography and surface functionalization techniques[9–11], which can utilize hydrophilic or charged surfaces to repel biomolecules or biocidal surfaces to inhibit microbial growth[12–14]. Despite significant progress in antifouling technologies, several challenges persist. First, conventional strategies exhibit limited mechanical robustness and stability as coatings may degrade or lose effectiveness over time due to environmental conditions or mechanical stress[15]. Second, antifouling coatings can sometimes obscure active sites on electrodes, rendering them inactive[16,17]. This often poses a considerable barrier to mass transport, impeding the diffusion of target analytes toward the sensor surface, which leads to reduced sensitivity and longer response times[17]. Third, the thickness is limited to a few tens of nanometers due to the potential adverse effects on the inherent performance of the electrode, making it challenging to achieve superior antifouling activities. Fourth, producing antifouling layers is costly, and thus scaling up for large-scale applications poses difficulties[18,19].

The manipulation of surface microarchitecture is a promising approach to achieving remarkable antifouling characteristics[20]. This method capitalizes on the intricate interplay between the small-scale physical features of a surface and the physicochemical properties of a fluid, thereby optimizing its hydrophobicity, capillary forces, and diffusion[21]. Notably, the implementation of porous structures, inspired by diverse biological systems, has proven effective in hindering biofouling through the control of pore size and capillary forces[22,23]. In addition to the antifouling capability, introducing interconnected pores into these coatings offers advantages that can enhance biomedical device functionalities by facilitating the efficient movement (i.e., diffusion) of fluids, ions, and molecules, enabling faster reaction kinetics and reduced response times[24]. This is particularly advantageous in applications such as biosensing (i.e., enhanced diffusion of molecules) that often utilize microfluidic systems where enhancing mass transport can improve signal detection, as well as in drug delivery (i.e., efficient release and distribution of therapeutic agents)[25,26]. Porous coatings also provide increased surface area for biodetection, which allows for greater interaction with biomolecules, such as nucleic acids, proteins, and cells, which enables a higher degree of biomolecular recognition and thus improves the sensing capabilities[26]. Surface packing density and film thickness are also critical factors for resisting non-specific biomolecular adsorption[27,28]. For example, a surface coverage of over 80% has been shown to be essential for maintaining stable resistance against non-specific adsorption[29], emphasizing the importance of high-density grafting of the surface layer.

We previously described development an albumin-based conductive nanocomposite formed via drop-casting in which the cross-linked protein matrix demonstrated excellent charge repulsion activity against non-specific molecules[16]. When this thin (~10 nanometer thick) porous antifouling coating was impregnated with electroconducting materials, such as gold nanowires (AuNWs) or graphene oxide flakes, there was significant enhancement of conductivity while maintaining good biofouling properties even in complex biological fluids, such as serum and plasma[16,30]. However, thin films with less than 50 nm thickness can face durability challenges over time due to physical shear stress[29,31], which could impede antifouling activities and potentially introduce interference in electrical signals. Also, because this thin antifouling coating was deposited using drop casting, it was applied to the entire surface of a multi-electrode array, which potentially can compromise the characteristics of the reference and counter electrodes and lead to a decrease in detection reliability[32]. For example, because the working electrode is the site where the desired electrochemical event occurs during molecular detection, the presence of an antifouling coating containing conductive materials at all sites could result in signal leakage between the electrodes and hinder the faradaic process at the working electrode (Supplementary Fig. 1).

In this work, we develop a nozzle-printing method to deposit an emulsion-templated, porous, albumin nanocomposite coating impregnated with conductive materials with increased thickness (~1 μm) in precise positions on the surface of a multiplexed gold electrode array and compare its functionality to the drop cast nanocomposite antifouling coating[16] that is approximately 100-times thinner (~10 nm). Both nanocomposite coatings consist of the same cross-linked bovine serum albumin (BSA) matrix containing gold nanowires (AuNWs) to enhance electron transfer to the underlying electrode. However, the nozzle-jet printing approach allow us to locally deposit the thicker emulsion coating on the working electrode without compromising the characteristics of the reference and counter electrodes. As a result, we are able to produce a thick porous coating with unparalleled antifouling and conductive properties that greatly enhances multiplexed electrochemical sensor sensitivity. We also demonstrate the increased capabilities of this approach by carrying out simultaneous detection of multiple clinically relevant bioanalytes—severe acute respiratory syndrome coronavirus 2 (SARS-CoV-2) nucleic acid, antigen, and host antibody—in clinical specimens with high sensitivity and specificity.

## Results
### Emulsion formulation and rheological properties
Porous structures can be prepared using various methods, including replica techniques, emulsion templating, direct foaming, capillary suspensions, and additive manufacturing[33,34]. However, we pursued the use of emulsion templating because it allows for the simple and fine control of morphology by manipulating phase states, droplet size, and packing density[24]. Additionally, optimizing the type of immiscible liquids and the degree of droplet dispersion can improve rheological properties, which can lead to improved process efficiency.

We chose to apply the emulsion to the surface of the electrodes using nozzle printing because it is a high-resolution and uniform patterning technique that offers several advantages over conventional printing techniques, such as screen-printing, drop-casting, and blade coating[35–37]. This approach not only reduces chip-to-chip variation but also ensures continuous processing, low-cost, and high-throughput processability[38,39], which can be crucial for future commercial scale-up. For the nozzle printing used here, we prepared an oil-in-water emulsion by ultrasonicating two immiscible liquids: an oil phase (hexadecane) and a water phase (phosphate buffer saline containing BSA and AuNWs) (Fig. 1a). To physically stabilize the matrix, glutaraldehyde (GA) was added to the emulsion immediately before printing, and then the emulsion was heated after printing to initiate cross-linking of the BSA[30,40] and promote oil evaporation, resulting in formation of interconnected nanoscale pores within a structurally stabilized matrix (Fig. 1a).

To monitor the maintenance of the emulsion state, we analyzed the size distribution of oil droplets in the emulsion by varying the sonication time. Dynamic light scattering (DLS) results indicated that the average droplet size decreased from 579.9 nm to 325.2 nm as the sonication time increased from 1 min to 25 min (Fig. 1b). When the sonication duration was extended to 40 min, the droplet size expanded back to the micron-scale. Notably, the sonication time of 25 min yielded a narrow size distribution, as characterized by a polydispersity index (PDI) value of 0.165.

To assess the shelf-life of the emulsion, UV–vis absorbance was measured (Fig. 1c) as the presence of oil droplets causes the scattering of light[41,42]. When the sonication time deviated from 25 min, phase separation between oil and water occurred within 30 min, resulting in reduced absorbance retention ratios below 90%. In contrast, sonication of the emulsion for 25 min resulted in remarkable stability, with a

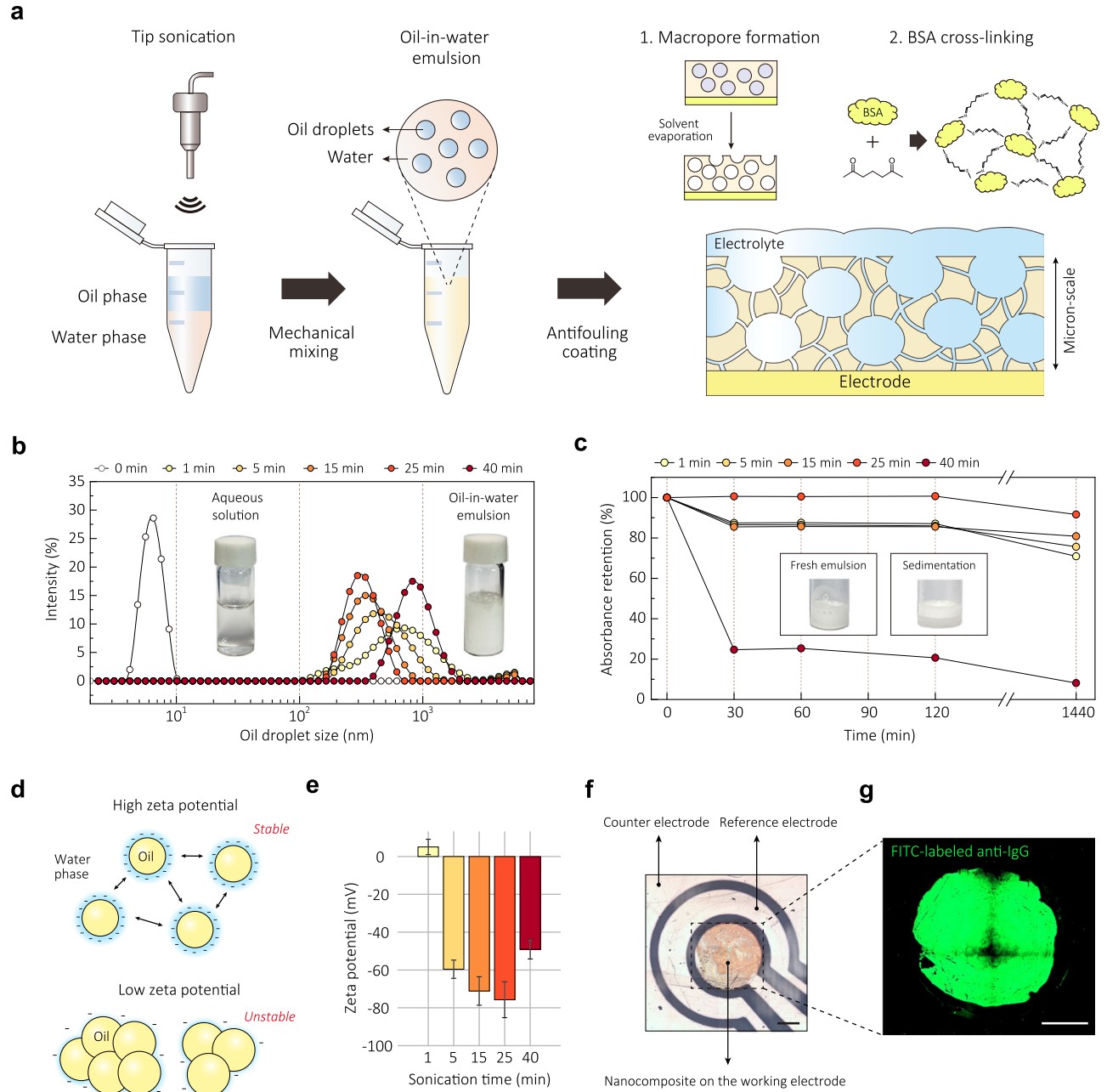

**Fig. 1 | Preparation of emulsion using ultrasonication and their rheological properties. a** Schematics of ultrasonication-based oil-in-water emulsion preparation and antifouling coating. BSA cross-linking and pore formation result in the development of a highly-interconnected porous nanocomposite with a micron-scale thickness. **b** Analysis of the oil droplet size distribution at various sonication time via DLS. The sonication time of 25 min yielded average droplet size of 325.2 nm with PDI value of 0.165. Inset images show visible difference between the aqueous solution and the oil-in-water emulsion. **c** Absorbance retention of emulsion with various sonication time. The absorbance values were obtained from UV–vis measurement at 280 nm. Emulsion sonicated for 25 min maintained 100% absorbance level until 120 min and retained 90% absorbance after a day. Inset images show fresh emulsion (left) at 40 min of sonication and their sedimentation (right) after 30 min of storage. **d** Illustration of correlation between surface charge of droplet and emulsion stability. Increasing surface charge leads to a stronger electrostatic

repulsive force between the droplets, preventing flocculation and maintaining droplet size. **e** Zeta potential values of emulsion with various sonication time. Sonication for 25 min yielded a high zeta potential value of −75.5 ± 9.5 mV. Data represents mean zeta potential with error bars indicating zeta deviation. This deviation was calculated based on bin values from each zeta potential distribution. **f** Precise patterning of emulsion on the working electrode, resulting in the formation of a uniform porous nanocomposite. The average diameter and center-to-center distance of the printed nanocomposite were measured as 1.52 ± 0.017 mm with coefficient of variation (CV) of 1.1% and 3.46 ± 0.091 mm with CV of 2.64%, respectively. **g** Confocal microscopy image of porous nanocomposite immobilized with FITC-labeled anti-IgG at excitation wavelength of 488 nm. The CV of intensity within the nanocomposite was 7.81%. Data reproducibility was confirmed by two independent experiments. Scale bars are 500 μm (**f**, **g**).

100% absorbance level being maintained for 2 h and 90% absorbance was retained even after a day. When the oil droplet diameter is less than 500 nm, nanoscale emulsions are less susceptible to gravitational separation and other physical forces, preventing undesirable

phenomena like flocculation and sedimentation over time[43]. These findings support the notion that reducing the size and improving the uniformity of oil droplets contributed to an extended shelf-life of the emulsion we observed.

Zeta potential offers information about the surface charge of dispersed droplets, further providing an indication of emulsion stability[44]. Increasing surface charge leads to a stronger electrostatic repulsive force between the droplets (Fig. 1d), preventing flocculation and maintaining droplet size[45]. Sonication for 25 min yielded a significantly high (absolute) zeta potential value of $-75.5 \pm 9.5$ mV (Fig. 1e and Supplementary Fig. 2). In contrast, deviations from this optimal sonication time decreased zeta potential. This observation is consistent with the results in Fig. 1b,c, confirming that a stable emulsion has a high zeta potential, while an unstable emulsion has a zeta potential approaching zero. Therefore, we optimized the sonication time to 25 min for subsequent experiments (Supplementary Fig. 3).

We also conducted a computational fluid dynamics (CFD) simulation to investigate the velocity field during nozzle printing (Supplementary Fig. 4b). For an aqueous solution without oil additives (control), inviscid flow resulted in a velocity surge at the nozzle tip, inducing drop splitting and unstable liquid ejection. In contrast, the emulsion displayed a moderate flow at the nozzle end, enabling stable patterning onto the electrode surface. The emulsion exhibited shear-thinning behavior, a non-Newtonian property where the viscosity decreases under high shear rates (Supplementary Fig. 4a). This behavior was modeled using the Carreau model, where the viscosity of the emulsion reached an appropriate value in the high-stress range[46]. The flow rate of emulsion at the nozzle tip was significantly reduced compared to the control solution, which exhibited Newtonian behavior with constant viscosity. As corroborated by the previous simulation[41], the emulsion demonstrated superior performance in nozzle printing by virtue of its shear-thinning behavior and the prevention of high-speed droplet splitting. Therefore, the emulsion formulation enables the application of nozzle printing that was unattainable using a BSA-based aqueous solution.

The capability to achieve uniform nanocomposite formation through nozzle printing is important because it allows for precise patterning of the emulsion on the working electrode and not on neighboring reference or counter electrodes (Fig. 1f, g and Supplementary Fig. 5). Importantly, the nozzle printing showed high reproducibility with electrode-to-electrode variation of 5.3% and chip-to-chip variation of 5.37% (Supplementary Fig. 7). We also analyzed the shelf-life of the devices under various storage conditions and confirmed that the sensor, stored at 25 °C in a nitrogen atmosphere, maintains high performance (Supplementary Fig. 8).

## Characterization of the emulsion-based nanocomposite

To explore whether the thick nozzle printed coating obtained with the optimized emulsion exhibits improved properties, we compared it to two different antifouling conducting coatings also composed of cross-linked BSA containing the same concentration of AuNWs: a thin (~10 nm) nanocomposite coating that was created with the drop casting method previously shown to show excellent antifouling properties[16] and a thicker coating with the same composition and thickness as the emulsion coating, which was fabricated using spin coating (Fig. 2a). SEM images confirmed that all three nanocomposites exhibited dense coatings with the corresponding thicknesses (i.e., ~10 nm for thin nanocomposite and 1 μm for both thick coating), although the emulsion exhibited larger pore sizes (17–51 nm vs 0.72–1.73 μm) (Fig. 2b).

The electrochemical characteristics of the nanocomposites at the interface with the solution were evaluated using cyclic voltammetry (CV) (Fig. 2c). Each nanocomposite underwent oxidation and reduction cycles in an electrochemically active solution. As expected, the cross-linking of albumin formed mesopores within the thin 10 nm coating, leading to relatively high peak currents. However, simply increasing the thickness of this nanocomposite to the micrometer scale using spin coating rendered the electrode nearly inactive, likely because of relatively decreased porosity (Fig. 2b). This thick

nanocomposite only displayed 5.73% oxidation and 12.4% reduction peak currents compared to a bare electrode. In contrast, the nozzle printed emulsion-based nanocomposite of similar thickness incorporated more pores that were larger (34.8 nm vs 1.13 μm), and this was accompanied by significantly enhanced electrochemical activity. The emulsion-based coating exhibited 59.2% oxidation and 63.5% reduction currents based on a diffusion-limited process despite being more than 1 μm thick (Fig. 2d). This improved performance also may be due to the presence of the AuNWs, which exhibit structural integrity within the composite, being more exposed to the solution through the macropores, thereby promoting nanoparticle-mediated electron transfer through the otherwise poorly conducting BSA matrix (Fig. 2e, Supplementary Fig. 9, 10).

To better understand the enhanced electrochemical properties of the porous emulsion-based nanocomposite, we conducted structural analyses. The BSA protein undergoes cross-linking through GA even in the presence of oil droplets as indicated by the UV–vis absorbance spectra, and this likely strengthens the structure before the formation of macropores (Supplementary Fig. 11). This is further evident from the negative time-of-flight secondary ion mass spectrometry (TOF-SIMS) spectrum of the nozzle printed coating, which revealed that $CN^-$ and $CNO^-$ signals were presented in its spectrum, indicating an abundance of the peptide backbone of BSA within the coating (Fig. 2f and Supplementary Fig. 12). Raman spectroscopic analyses also revealed a shift of the C-N peak, confirming the formation of peptide bonds between BSA and GA (Supplementary Fig. 13). Atomic force microscopy (AFM) and mercury intrusion porosimetry (MIP) measurements verified the presence of macropores with an average size of 1.123 μm and mesopores with an average size of 9.53 nm (Fig. 2g, h). In addition, the emulsion-based nanocomposite exhibited a 38.7-fold increase in Brunauer−Emmett−Teller (BET) surface area compared to the thin nanocomposite (Supplementary Fig. 14). These analyses confirmed the successful fabrication of a thick and highly porous nanocomposite composed of cross-linked BSA containing AuNWs. The engineered porosity effectively addresses the challenge of electrochemical deactivation that compromises many conventional antifouling coating approaches.

## Enhancement of electrochemical performance and antifouling activities

Field-deployable diagnostics must be portable and cost-effective, and provide accurate results rapidly using convenient sample types such as saliva or nasal swabs[47]. To compare the electrochemical performance and antifouling activities of the three antifouling conductive coatings, we developed biosensors for detection of the SARS-CoV-2 gene, ORF1a, using the CRISPR/Cas12a-based nucleic acid detection approach in which Cas12a becomes activated in the presence of a target RNA and cleaves a single strand DNA (ssDNA) reporter probe conjugated with biotin that is normally bound to a complementary protein nucleic acid (PNA) immobilized on the surface of the nanocomposite. The presence of bound biotinylated reporter probe is detected by addition of poly streptavidin-horseradish peroxidase (HRP) and addition of a precipitable form of its substrate, 3,3′,5,5′-tetramethylbenzidine (TMB), which results in its localized deposition on the surface of the working electrode[48] and an increase in peak current. Thus, the detection of the target RNA results in CRISPR-based cleavage of the bound reporter probe and a reduction in peak current (Fig. 3a).

To examine the diffusion phenomena of electroactive species across the nanocomposite structures, we carried out CV using potassium ferro/ferricyanide solution prior to PNA immobilization on the electrodes (Fig. 3b, left). In line with the results displayed in Fig. 2c, d, the thin nanocomposite exhibited a higher peak current than the thick emulsion-based nanocomposite. This can be attributed to the enhanced diffusion of electroactive species owing to the nanoscale

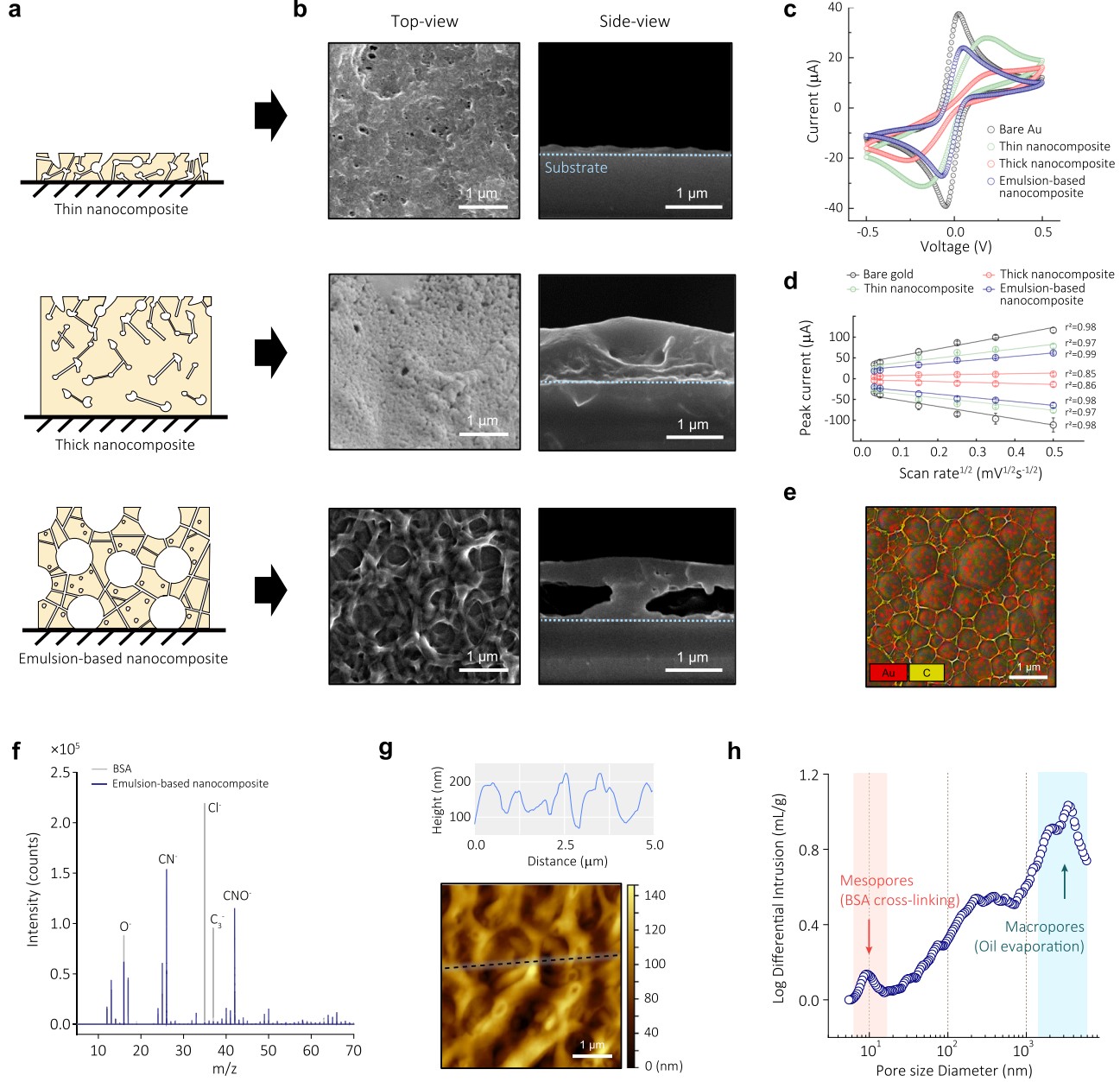

**Fig. 2 | Characterization of cross-linked nanocomposites. a** Schematic of three different cross-linked BSA nanocomposites: thin, thick, and porous emulsion-based nanocomposites. **b** Top- and side-view SEM images of nanocomposites. Three nanocomposites exhibited dense coatings with corresponding thicknesses and porosities. Data reproducibility was confirmed by three independent experiments. **c** Representative CV data in 5 mM ferri-/ferrocyanide solution with three nanocomposites and bare Au electrode. Scan rate is 0.1 V s⁻¹ between −0.5 V and 0.5 V. **d** Oxidation and reduction peak current versus the square root of scan rate (from 0.07 to 1.0 V s⁻¹). Data represents as mean values ± SD (*n* = 4 independent experiments). **e** Energy dispersive analysis (EDS) map for Au and C in emulsion-based nanocomposite. **f** Negative TOF-SIMS spectra with emulsion coating. *y*-axis denotes the detected number of secondary ions and *x*-axis denotes the mass-to-charge (*m/z*) ratio. CN⁻ and CNO⁻ signals were predominantly presented, indicating abundance of the peptide backbone in the emulsion-based nanocomposite. **g** AFM topography of porous nanocomposite (bottom) and the extracted height profile of black dotted line (top). Data reproducibility was confirmed by two independent experiments. **h** Measurement of porosity via MIP. Emulsion-based nanocomposite has mesopores with an average size of 9.53 nm between the macropores with an average size of 1.123 µm.

thickness of thin nanocomposite. The thick spin-coated nanocomposite produced a weak current, rendering it unsuitable for use in electrochemical sensors, and thus, we did not further explore this coating. Importantly, the thick emulsion-based coating demonstrated the most reproducible peak currents, showing the potential to create a more reliable sensing platform.

When we analyzed the currents of the three nanocomposite-coated sensors using this approach, we found that the thick emulsion-based nanocomposite delivered more than 2.3-fold higher peak current than the thin nanocomposite (Fig. 3b, right). This result suggests an increased density of PNA and bound reporter probe as well as resultant TMB precipitation, which is likely driven by the expanded surface area available for redox reactions. This initial high peak current is pivotal for ensuring high sensitivity in CRISPR-based electrochemical diagnostics. During the analysis of confocal microscopy using FITC-labeled anti-IgG, we were able to further validate this trend by observing a 4.41-fold increase in fluorescent intensity in the emulsion-based nanocomposite compared to the thin nanocomposite (Supplementary Fig. 15).

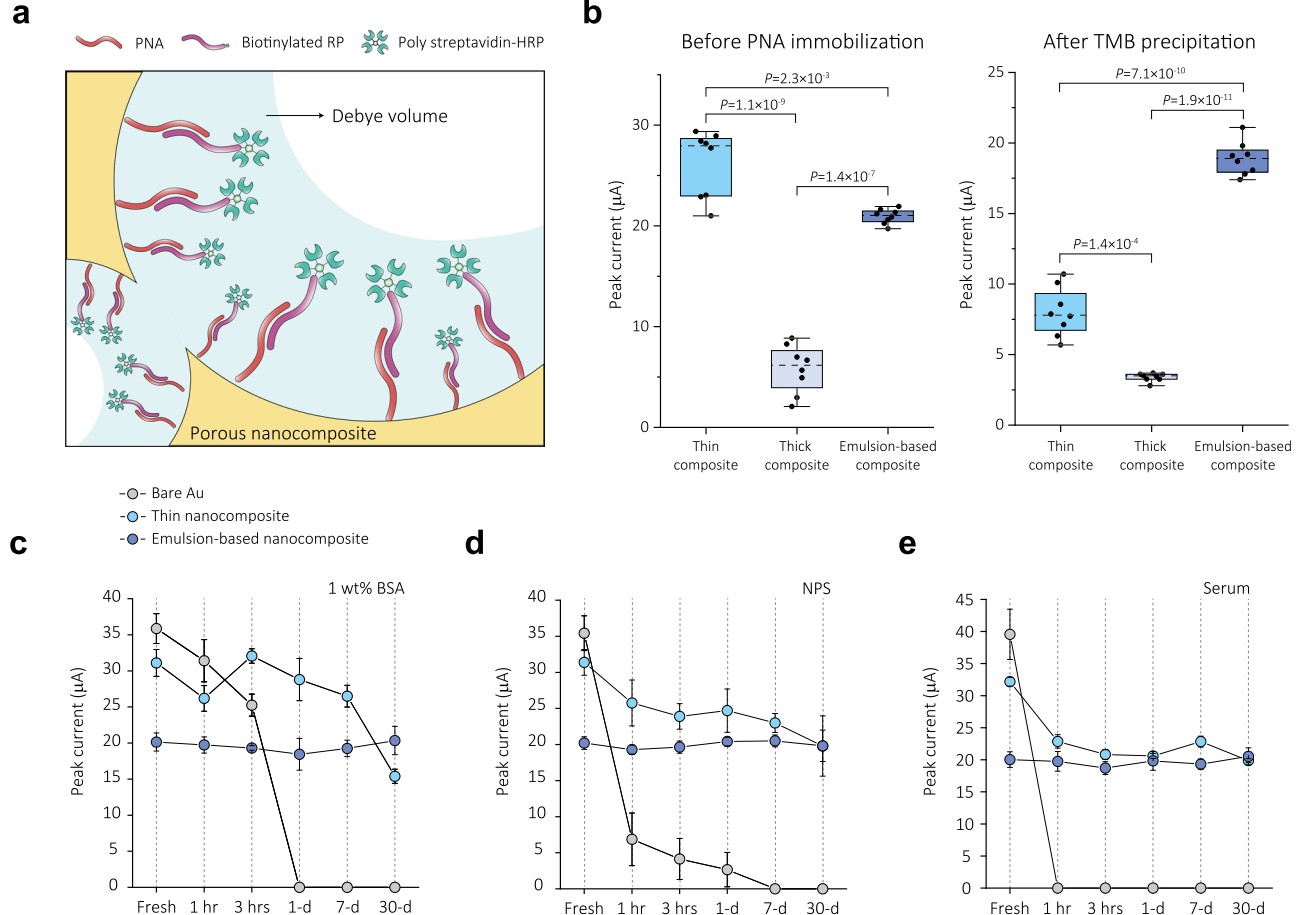

**Fig. 3 | Enhancement of electrochemical performance and antifouling activities by emulsion-based nanocomposite. a** Fabrication of CRISPR/Cas12a-based electrochemical nucleic acid sensor by immobilizing PNA/reporter probe (RP)-HRP onto porous nanocomposite. Debye volume can be maximized in concave structures. **b** Measurement of CV according to nanocomposite structures. CV were conducted in 5 mM ferri-/ferrocyanide solution before PNA immobilization (left). After PNA immobilization, RP-HRP hybridization, and TMB precipitation, CV were conducted in PBST (right). Scan rate is 0.1 V s⁻¹ between −0.5 and 0.5 V. Statistical significance was tested using two-tailed Student's *t* test. Two chips were used for the measurements. For all boxes, the dashed central line represents the median, the bottom and top edges mark the 25th and 75th percentiles. The whiskers denote 1.5× interquartile range (IQR)(*n* = 8 independent experiments). **c**−**e** Comparison of peak current between bare Au electrode and nanocomposite-coated electrodes. Chips were stored for one month at 4 °C in 1% BSA, NPS, and serum. Data represents mean values ± SD (*n* = 4 independent experiments). Emulsion-based nanocomposite-coated electrodes demonstrated superior antifouling properties against non-specific molecules by maintaining its electrochemical behavior for one month under all biofluid conditions.

We also evaluated the robustness of the nanocomposite-coated sensors in the face of various biofluids (Fig. 3c-e and Supplementary Fig. 16). When chips were immersed in a 1% BSA solution or complex clinical samples, such as serum and nasopharyngeal samples, the bare Au electrodes quickly lost their activity due to the continuous adsorption of non-specific molecules and conductivity was entirely lost within 1 h in serum. The thin nanocomposite-coated electrodes exhibited relatively stable electrochemical characteristics for up to a week in a 1% BSA solution, consistent with past studies[16]. However, when subjected to these complex clinical samples, they experienced a steady current decrease, with approximately a 28% compared to the initial peak current after a week of exposure for both samples. In contrast, electrodes coated with the thick emulsion-based nanocomposite demonstrated superior antifouling properties against non-specific molecules by maintaining its electrochemical behavior for an entire month under all biofluid conditions (Fig. 3c−e). Compared to the initial peak current of approximately 20 μA, they exhibited only a slight signal decrease of about 1.9% over the course of a month and showed a high device uniformity with a coefficient variation of 8%. This minimizes signal drift and maintains a stable baseline, enabling precise detection of target analytes within complex specimens (Supplementary Fig. 17). The emulsion-based nanocomposite also showed strong

structural robustness and stability, maintaining its performance under mechanical stress and minimizing signal drift (Supplementary Fig. 18).

## Electrochemical detection of viral infection

POC diagnostics play an indispensable role in detecting viruses beyond the confines of laboratories, thereby mitigating community transmission, as evidenced by the COVID-19 pandemic[49]. A comprehensive analysis of nucleic acid, antigen, and serological diagnosis, can significantly increase testing accessibility, and expedite containment and treatment strategies. We therefore explored whether we could develop a biosensor capable of simultaneous detection of SARS-CoV-2 RNA, antigen, and host antibody with enhanced sensitivity using the thick emulsion-based antifouling coating (Fig. 4a).

We used the CRISPR/Cas12a detection approach to detect the presence of SARS-CoV-2 RNA, which is measured as a decrease in peak current that is inversely proportional to the target RNA concentration. Reverse transcription and recombinase polymerase amplification (RT-RPA) were used to amplify the ORF1a gene of SARS-CoV-2 because RT-RPA can amplify dsDNA up to 1 kb under moderate conditions (i.e., at room temperature or 37 °C)[50] (Supplementary Table 1). Four sets of primers for RT-RPA were evaluated and optimized using a fluorescence readout (Supplementary Figs. 19, 20). To evaluate sensitivity in relation

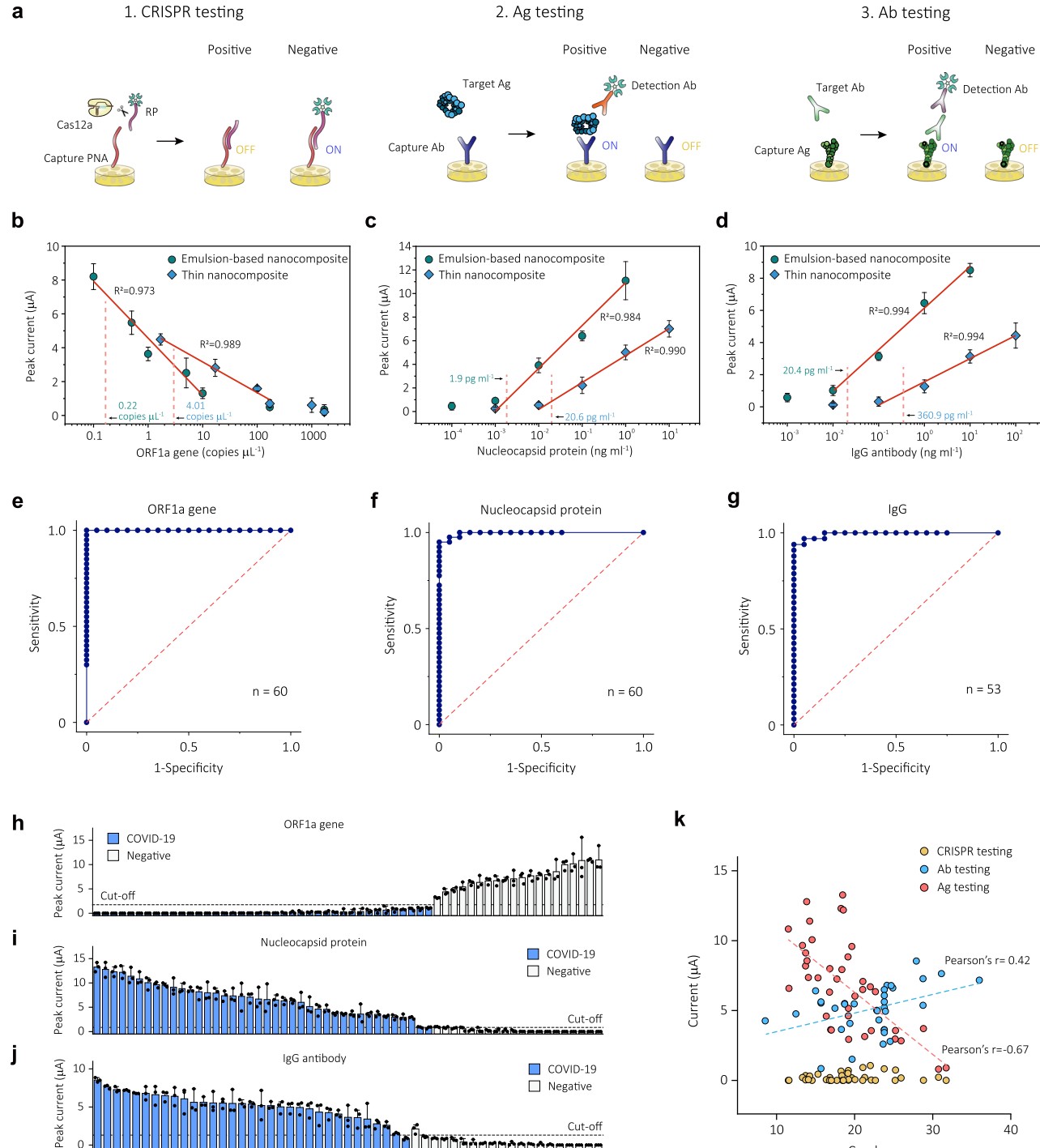

**Fig. 4 | Electrochemical detection of viral infection using emulsion-based sensor. a** Schematics of electrochemical enzymatic detection of SARS-CoV-2 RNA, antigen, and host antibody using emulsion-based sensors. **b–d** Calibration curves for ORF1a gene and nucleocapsid protein of SARS-CoV-2, and IgG antibody using CV with 1 V s⁻¹ scan rate between −0.5 and 0.5 V. LOD was defined using three standard deviations (3σ) of the blank solution. CV were measured on four WEs, out of which three were involved in the reaction with the target species, while one served as a negative control. Data represents as mean values ± SD (*n* = 3 independent experiments). **e–g** ROC curves based on the detection results of emulsion-based sensors. AUC was 1, 0.996, and 0.993 for ORF 1a, nucleocapsid protein, and IgG antibody, respectively. Clinical NPS (Positive: 40, Negative: 20) were used to

detect ORF1a gene and nucleocapsid protein of SARS-CoV-2, and clinical serum (Positive: 33, Negative: 20) was used to detect the IgG. Each data obtained from three independent electrodes. The experiments were conducted over a total of two rounds. **h–j**, Waterfall distribution of peak current for clinical samples. Data represents mean values ± SD (*n* = 3 independent experiments). Cut-off values were determined from the ROC curves: 2.12 (ORF1a gene), 0.857 (Nucleocapsid protein), and 1.3 (IgG antibody) μA. **k** Correlation between peak current measured from emulsion-based sensors and $C_t$ value measured from RT-qPCR for COVID-19 positive clinical samples. Pearson's r was −0.67 for antigen testing and 0.42 for antibody testing.

to the nanocomposite structures, we incubated the RT-RPA product, Cas12a/gRNA, and reporter probe on electrochemical sensors with different antifouling coatings and measured the peak current after the precipitation of TMB. The calibration curve of the thin nanocomposite-coated sensor displayed a linear increase from 1.7 to 170 copies $\mu l^{-1}$, with a limit of detection (LOD) of 4.01 copies $\mu l^{-1}$ (Fig. 4b). The calibration curve of the thick emulsion-based nanocomposite sensor demonstrated a linear increase in peak current from 0.1 to 10 copies $\mu l^{-1}$, marking a sensitivity improvement of over 3.5-fold (LOD of 0.22 copies $\mu l^{-1}$), which is based on the enhanced electrochemical performance of the sensor.

In addition to the CRISPR-based detection, we implemented an affinity-based sandwich strategy into the thick emulsion-based sensor, broadening its detection capability to include both SARS-CoV-2 antigen and host antibody[51] (Supplementary Figs. 21–23). This strategy involves capturing the target antigen and antibody respectively using an antibody and antigen that are immobilized on the surface of the nanocomposites. Subsequently, these targets interact with a biotinylated secondary antibody, which binds with HRP and supports TMB precipitation. The optimization process for this sensor with the thick emulsion-based coating that is tailored for immunological diagnostics is presented in Supplementary Figs. 24, 25. Notably, this sensor exhibited a significantly improved sensitivity to the SARS-CoV-2 nucleocapsid protein, displaying a LOD of 1.9 pg ml$^{-1}$, which is 10-fold lower than that of the thin nanocomposite (Fig. 4c). Similarly, marked linearity was observed for IgG detection, with a relative LOD improvement of 17-fold (Fig. 4d). These data demonstrate the ability of the thick porous emulsion-based coating to substantially enhance target protein sensitivity while reducing noise, owing to its exceptional antifouling properties.

We then assessed the performance of biosensors coated with the thick porous nanocomposite using clinical samples. We initially conducted electrochemical detection of the ORF1a gene with these sensors using nasopharyngeal samples obtained from patients (Positive: 40, Negative: 20) (Supplementary Table 2). We also used the sensors to detect SARS-CoV-2 nucleocapsid protein in the same samples (Positive: 40, Negative: 20) and IgG in serum (Positive: 33, Negative: 20), respectively (Supplementary Tables 3, 4). The diagnostic results were deduced from the observed peak currents (Supplementary Fig. 26). The receiver operating characteristic (ROC) curves for all targets illustrated impressive diagnostic accuracy with area under the curve (AUC) values of 1 for ORF1a, 0.996 for the antigen, and 0.993 for the antibody (Fig. 4e–g). The ROC curves also provided cut-off current values that optimized the sum of sensitivity and specificity: 2.12 (ORF1a), 0.857 (antigen), and 1.3 (IgG) $\mu A$. Utilizing these cut-off values, the emulsion-based nanocomposite biosensor accurately differentiated between positive and negative clinical samples with high sensitivity and specificity (Fig. 4h–j and Supplementary Table 5). The current results demonstrate the performance of biosensors coated with the thick emulsion-based antifouling nanocomposite across various clinical diagnostic applications.

We further explored the correlation between the electrochemical detection results and cycle threshold ($C_t$) values from RT-qPCR for the clinical samples (Fig. 4k). The plot revealed that the peak current from nucleic acid detection maintained relative consistency to the changes in $C_t$ values. The CRISPR/Cas12a-based sensor was successful in detecting ORF1a even at high $C_t$ values (>30). Moreover, a negative correlation was identified between $C_t$ values and antigen detection levels with a Pearson's r value of −0.67. This suggests that the detected antigen increases in tandem with the viral load in the clinical sample, which is consistent with the clinical observation that SARS-CoV-2 RNA and antigen levels typically rise proportionally during the initial 6 days post-infection and subsequently decrease[52,53]. Meanwhile, a mild positive correlation between $C_t$ values and IgG detection levels was observed, reflected by a Pearson's r value of 0.42. Unlike RNA and

antigen, antibodies begin to be detected after 6 days of viral infection due to seroconversion, and their titers remain stable for several months, echoing our antibody detection results[52,53]. The remarkable AUC values achieved for ORF1a, the antigen, and the antibody underscore the high sensitivity and specificity of the emulsion-based multiplexed biosensor, suggesting its practical efficacy for diagnostics of viral infections.

## Multiplexed detection of viral RNA, antigen, and antibody

The complex nature of COVID-19 infections requires a comprehensive approach to analyzing test results for effective patient management and pandemic control, which can include molecular, antigen, and serology tests[49]. However, there is an unmet need for advanced diagnostic technologies that can carry out all of these detection assays simultaneously, which could facilitate a more accurate and comprehensive assessment of viral infections[54]. For this reason, we set out to simultaneously detect SARS-CoV-2 nucleic acid, antigen, and host antibody by spotting different capture probes onto adjacent working electrodes within the same sensor chip using the precise control capability of our nozzle jet printing method. Previous research suggests that nasopharyngeal specimens from severe SARS-CoV-2 patients contain detectable levels of RNA, antigen, and antibodies[55]. However, for the patients we examined were not as severely affected, and IgG was not detected in their sample across a broad spectrum of $C_t$ values. Therefore, we spiked patient serum into the nasopharyngeal samples to verify the performance of our multiplexed COVID-19 biosensor. We conducted two consecutive assays: initially, 20 μl of the serum-spiked nasopharyngeal sample was incubated on a chip to detect both nucleocapsid protein and host antibody, and then RT-RPA was carried out using 5 μl of the sample, and the resulting products were incubated on the same chip with Cas12/gRNA and reporter probe. Each target was then quantified simultaneously as an electrochemical signal, facilitated by the binding of poly streptavidin-HRP and the precipitation of TMB.

Four experimental sets were designed to evaluate the performance of multiplexing (Fig. 5a, b). Each set consisted of four distinct combinations determined by the presence or absence of spiked antibodies in COVID-19 positive and negative patient samples. In the positive sample spiked with IgG (SET 1), a positive molecular diagnostic response was observed for ORF1a and both nucleocapsid protein and IgG targets also were detected as the difference between positive samples and the negative control is statistically significant. Conversely, in COVID-19 positive nasopharyngeal samples without spiked IgG (SET 2), there was no significant alteration in the peak current from IgG detection as expected. Notably, the three targets were successfully distinguished with 100% accuracy in COVID-19 negative clinical samples based on the presence or absence of IgG (SET 3 and 4). These results suggest that the excellent antifouling properties and high porosity of the emulsion-based nanocomposite biosensor can resist non-specific binding and accurately recognize several target molecules simultaneously with excellent sensitivity and specificity in human clinical samples. We anticipate that electrochemical sensors coated with the emulsion-based antifouling coating could be used to diagnose SARS-CoV-2 infection across a broader temporal range and to monitor disease progression. They may also offer potential utility in evaluating the efficacy of vaccination responses by simultaneously detecting viral RNA, antigen, and antibodies.

## Discussion

In this study, we developed a nozzle printing method for selectively templating an emulsion-based, micrometer thick, porous coating that has both excellent antifouling and electroconducting properties on the surface of working electrodes, but not over closely apposed reference and counter electrodes (Fig. 6a). The technology leverages the unique properties of oil-in-water emulsions to achieve precise

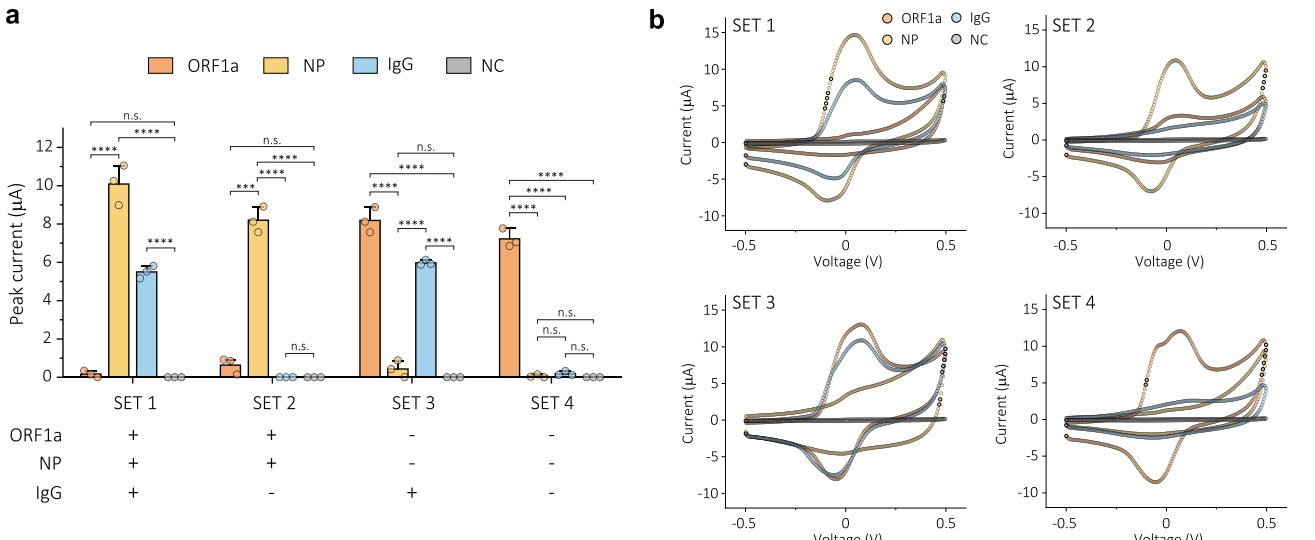

**Fig. 5 | Multiplexed detection of viral RNA, antigen, and antibody using emulsion-based sensor. a** Histogram showing the multiplexing performance of emulsion-based sensors for SARS-CoV-2 ORF1a, nucleocapsid protein (NP), IgG antibody, and negative control (NC). Four experimental sets are combinations of COVID-19 positive and negative nasopharyngeal samples based on the presence or absence of spiked IgG. Data represents as mean values ± SD ($n = 3$ independent electrochemical chips). Statistical significance was tested (***$P < 0.001$, ****$P < 0.0001$; two-tailed Student's $t$ test). **b** Representative CV data of multiplexed sensors for SARS-CoV-2 ORF1a, NP, IgG antibody, and NC. Three chips were used for each set of experiments.

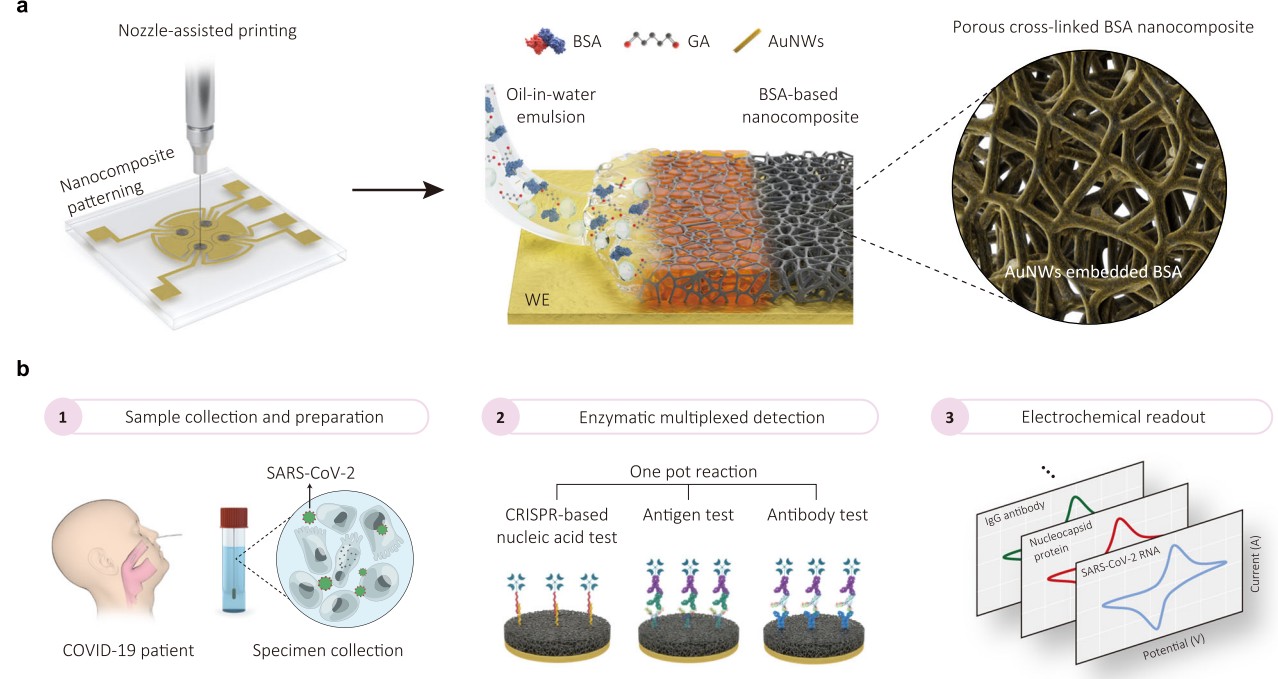

**Fig. 6 | Thick and porous antifouling nanocomposite for electrochemical detection of virus with high accuracy and reliability. a** Fabrication of emulsion-processed porous antifouling nanocomposite via nozzle-assisted printing. The AuNWs are embedded to the nanocomposite when BSA is cross-linked by GA. Electrochemical sensor consists of four working electrodes (WE). **b** Overview of multiplexed detection of SARS-CoV-2 RNA, antigen, and host antibody using emulsion-based nanocomposite electrochemical sensor.

control over droplet size, surface charge, and ink stability. The removal of oil components within the composite results in a uniform distribution of pores, which leads to synergistic antifouling effects at the micro-scale level and enhanced diffusion through interconnected pores. The porous nanocomposite surface also can be easily functionalized with multiple capture probes via carbodiimide reaction, enabling enzymatic recognition of multiple target molecules[56]. This

nanocomposite coating successfully mitigates the challenges of bio-fouling even with complex biological fluids and probe loading, and thus enhances the diagnostic performance of electrochemical sensors, as well as facilitates their multiplexing on the same chip (Supplementary Table 6). Impressively, because of its porosity and the presence of conducting AuNWs within the cross-linked BSA nanocomposite, the coating maintains efficient electron transfer despite

its thickness exceeding one micrometer (i.e., 100 times thicker than conventional antifouling matrices). Because of the interconnected pores achieved through BSA cross-linking and oil evaporation from the emulsion, the integration of nanoelectrodes promote enhanced electron diffusion, and thereby proper signal transmission at the working electrode.

A key element of our fabrication method is the use of nozzle printing, which is a high-resolution and uniform patterning technique that offers several advantages over conventional printing techniques such as screen-printing, drop-casting, and blade coating[35–37]. This approach not only reduces chip-to-chip variation but also ensures continuous processing, low-cost, and high-throughput processability[38,39], which can be crucial for future commercial scale-up. Using this method, we fabricated multiplexed electrochemical sensors that enabled simultaneous detection of SARS-CoV-2 RNA, antigens, and host antibodies with high sensitivity and specificity (Fig. 6b). The porous nanocomposite surface can be easily functionalized with multiple capture probes via carbodiimide reaction, enabling enzymatic recognition of multiple target molecules[56]. The exceptional antifouling activity of nanocomposite prevents signal degradation from non-specific species in nasopharyngeal secretions and serum. This advantage presents the potential to streamline sample pre-processing steps in on-site diagnostics. The developed high-precision diagnostic technology enables the collection of extensive immunological data in a simplified manner, allowing for a deeper understanding of the correlation between biomarkers of virus infection. Moreover, it has the potential to contribute to the analysis of individual immunity and vaccine efficacy, thereby improving quarantine measures and individual-tailored medical strategies and enabling prompt response to future infectious diseases. In addition to virus detection, we have also confirmed the robust and reproducible detection of other biomarkers using the emulsion-based sensor. The sensor exhibited high sensitivity in detecting anti-glucose regulating protein 78 (anti-GRP78) and anti-protein kinase R-like endoplasmic reticulum kinase (anti-PERK), which are crucial indicators for cancer, diabetes, and rheumatoid arthritis, demonstrating its potential in diverse disease diagnostics (Supplementary Fig. 27).

This coating technology holds significant promise in the field of electrochemical biosensors. The excellent antifouling activity of the coating effectively prevents signal degradation caused by non-specific species in complex biofluids, which can streamline sample pre-processing steps in on-site diagnosis, simplifying overall testing methodology and reducing the occurrence of false signals. Furthermore, the ability to functionalize the surface of the porous nanocomposite with multiple capture probes allows for the enzymatic recognition of various target molecules, including viral RNA, antigens, and host antibodies, as we demonstrated here. This comprehensive diagnostic ability could be vital in managing future pandemics, where swift and accurate diagnosis is crucial. The collection of extensive diagnostic data in a simplified manner should enhance our understanding of the correlations among various biomarkers over the course of a viral infection. This may contribute not only to the enhancement of diagnostic accuracy and the monitoring of viral outbreaks, but also to our comprehension of disease progression and the customization of patient management strategies. Finally, the creation of a robust barrier against non-specific adsorption presents opportunities for other types of biomedical devices as well, including healthcare monitoring and implantable devices. By reducing undesirable interactions with non-specific biomolecules, this technology has the potential to amplify device performance, durability, and potentially mitigate inflammatory responses, thereby improving their reliability.

## Methods
### Preparation of electrochemical chip
Electrochemical chips with working, reference, and counter electrodes were fabricated by depositing metallic electrodes (Cr/Au = 5/50 nm)

onto a glass substrate (2.5 cm × 2.0 cm) using an e-beam evaporator (SNTEK Co., Ltd.). Before use, the chips underwent a cleaning process consisting of a 5 min sonication with acetone followed by another 5 min sonication with isopropanol. Subsequently, the chips were subjected to $O_2$ plasma treatment (Femtoscience Inc., Korea) at 80 W for 8 min.

### Emulsion preparation using ultrasonication
To prepare the emulsion, a solution of 10 mg ml$^{-1}$ bovine serum albumin (BSA) (Sigma Aldrich, USA, no. A7906) and 30% (v/v) gold nanowires (AuNWs) with diameter of 30 nm and length of 4.5 μm (Sigma Aldrich, USA, no. 716944) in PBS (Sigma-Aldrich, USA, no. D8537) was added to hexadecane (Sigma-Aldrich, USA, no. H6703) in a volume ratio of 2:1. The mixture was then sonicated using a micro-tip sonicator at 20% amplitude, with a pulsing pattern of the 30 s on and 10 s off, for a total duration of 25 min (VC 505, Sonics & Materials). After sonication, the resulting emulsion was mixed with 70% glutaraldehyde (GA) (Sigma Aldrich, USA, no. G7776) in a volume ratio of 70:1. Prior to this, the GA was diluted in PBS at a volume ratio of 1:7.

### Characterization of emulsion and porous nanocomposites
Emulsion were sonicated with 1, 5, 15, 25, and 40 min, and subsequently diluted to a 1:100 ratio for UV–vis spectroscopy, zeta-potential, and dynamic light scattering measurements. UV–vis spectroscopy (Lambda 1050, Perkin Elmer) was measured at 270 nm. Zeta potential and dynamic light scattering measurements (Zetasizer Nano, Malvern Instruments Ltd.) were measured at 25 °C. Contact angle measurements (SEO Phoenix) of the emulsion and an aqueous solution dissolved with 10 mg ml$^{-1}$ BSA were performed with $O_2$ plasma-treated gold chip. Porosity was measured by the mercury intrusion porosimetry (MIP) (Autopore 9605, Micromeritics). BET surface area was measured by the Nitrogen physisorption (3Flex, Micromeritics). BSA concentration was measured using Nanodrop (NanoDrop™ One, Thermo Fisher). Raman spectra and TOF-SIMS were measured using ARAMIS (Horiba Jobin Yvon) and TOF-SIMS5 (ION-TOF GmbH), respectively. Surface characterizations of both top and cross-sectional views were characterized using SEM (Hitachi S4800). Top views of all three nanocomposites were obtained by coating the samples onto $O_2$ plasma-treated glass substrates. Cross-sectional views of all three nanocomposites were carried out by coating the samples onto $O_2$-plasma treated Si wafers. The topography and height profile of the emulsion-composite was measured by AFM (Probes Co, LTD., Korea). The AFM sample was prepared by coating the emulsion-composite onto an $O_2$-plasma treated Si wafer.

### Rheology measurement and CFD analysis of nozzle printing
The rheology measure is done with MCR 302 rheometer (Anton Paar). Shear rate ($\dot{\gamma}$) was swept from 0.01 to 100 s$^{-1}$. Shear stress and viscosity are determined when steady stress is reached. Samples are freshly prepared before measurements. Emulsion is fitted to the Carreau model whose shear rate dependent viscosity is defined as previous article.

$$\mu_{app} = \mu_{inf} + \left(\mu_0 - \mu_{inf}\right)\left[1 + (\lambda\dot{\gamma})^2\right]^{\frac{n-1}{2}} \tag{1}$$

where $\mu_{inf}$ is infinite shear rate viscosity; $\mu_0$ is zero shear rate viscosity; $\lambda$ is relaxation time; and $n$ is power index. For control ink, Newtonian model is used whose viscosity is constant independently to $\dot{\gamma}$.

To study the velocity field of nozzle printing with two rheology models, computational fluid dynamic (CFD) simulation is conducted with COMSOL Multiphysics. Navier-stokes equations are used as governing equation to describe incompressible Newtonian fluids in geometry shown in Supplementary Fig. 3. Because the flow rate was in Laminar flow regime (Re=ρUL/μ < 10), the turbulent effect is not

included [1]. For liquids with viscosity $\mu$, momentum and mass conservation equations are solved as

$$\rho\left(\frac{\partial u}{\partial t} + u \cdot \nabla u\right) = -\nabla p + \nabla \cdot \mu\left(\nabla u + \nabla u^T\right) \quad (2)$$

$$\nabla \cdot u = 0$$

where u is the flow velocity, $p$ is the pressure. Liquid properties follow the rheological measurement. Equation (1) is coupled with apparent viscosity ($\mu_{app}$) defined by Carreau model in rheology measure section. The meshing is constructed under 'higher' option, which is tested confine enough.

## Fabrication of thin-, thick-, emulsion-based nanocomposites

The emulsion-processed nanocomposite was patterned onto the working electrode using a nozzle-assisted printer (BIO X6, CELLINK) at a pressure of 5 kPa and a printing speed of 20 mm s$^{-1}$. The printing bed was heated to 30 °C during printing. After printing, the chips were placed in an oven at 80 °C for 30 min to induce BSA crosslinking and evaporation of the oil droplets. Subsequently, the emulsion-processed nanocomposite was washed with PBS in a shaker at 400 rpm for 30 min to remove any residual oil, followed by DI-water washing and drying with N$_2$ blowing to eliminate any remaining chemicals.

For the thin-nanocomposite, an aqueous solution consisting of 10 mg ml$^{-1}$ BSA and 30% (v/v) AuNWs in 10 mM PBS was mixed with glutaraldehyde (GA) in a volume ratio of 68:2. The mixture was dropcasted onto the chip preheated to 85 °C for 30 s. The thin-nanocomposite was washed with PBS in shaker for 5 min, followed by DI-water washing and drying using N$_2$ gas. Thick nanocomposite was prepared using the same aqueous solution. The aqueous solution was spin-coated at 500 rpm for 45 s. The resulting liquid film was then heated in an oven at 85 °C for 30 min. The chip was washed with PBS in a shaker for 5 min, followed by DI-water washing and drying with N$_2$ gas.

## Electrochemical characterization of the nanocomposites

Before EC characterization, all nanocomposites were functionalized using a solution of 400 mM 1-ethyl-3-(3-dimethylaminopropyl) carbodiimide hydrochloride (EDC) (Sigma Aldrich, USA, no. E7750) and 200 mM N-hydroxysuccinimide (NHS) (Sigma Aldrich, USA, no. 130672) in a 50 mM MES buffer (Sigma-Aldrich, USA, no. M1317) for 30 min. To quench any unreacted functional groups, the nanocomposites were incubated with 1 M ethanolamine (Sigma-Aldrich, USA, no. 398136) in PBS for 30 min. Cyclic voltammograms (CV) were performed in 5 mM [Fe(CN)$_6$]$^{3-/4-}$ (Sigma-Aldrich, USA, no. P3289 and no. 702587) in 1 M KCl (Sigma-Aldrich, USA, no. 58221) with a scan rate of 0.1 V s$^{-1}$, covering a voltage range from −0.5 V and to 0.5 V (ZIVE SP1, WonATech, Co., Ltd.). The peak oxidation currents of all nanocomposites were calculated using IVMAN 1.5 software. To evaluate the antifouling activities, the chips were incubated in a 1 wt% BSA solution, nasopharyngeal specimens, and serum for various durations: 1 h, 3 h, 1 day, 1 week, and 1 month. Bare gold was used as a control to assess any decrease in electrochemical properties. The shelf-life of the emulsion-processed nanocomposite was tested under two different conditions: PBS at 4 °C and dry at 4 °C for various durations: 1 h, 1 day, 3 days, and 1 week.

## Fluorescence characterization of nanocomposites

All nanocomposites were functionalized using a solution of 400 mM EDC and 200 mM NHS in a 50 mM MES buffer for 30 min. Nanocomposites were spotted with FITC-labeled anti-IgG (Sigma-Aldrich, USA, no. F9512) of 1 mg ml$^{-1}$ and washed thoroughly with PBS. Fluorescence images were demonstrated with the confocal microscope (Andor Dragonfly 200) with an excitation wavelength 488 nm. Fluorescence intensities were calculated by ImageJ measurements.

## Reverse transcription-recombinase polymerase amplification (RT-RPA)

According to the instructions provided in the Twist Amp manual, the primers and probe for the RT-RPA were designed to amplify a specific segment of the SARS-CoV-2 ORF gene. The sequence of the primers (Bioneer, Korea) utilized for ORF gene detection is as follows: forward primer: 5'- AAATTGTTAAATTTATCTCAACCTGTGCTTGT-3', Reverse primer: 5'-AGTTTCTTCTCTGGATTTAACACACTTTCT-3'. RT-RPA was performed in a reaction mixture containing 2.4 µl each of forward and reverse primer (10 µM), 29.5 µL rehydration buffer, 2.5 µL SYBR green I dye (20x), 1 µl M-MLV reverse transcriptase (200 U µl$^{-1}$), 1.5 µl Murine Rnase inhibitor (40 U µl$^{-1}$), 5 µl target RNA, and 3.2 µl nuclease-free water. The mixture was vortexed and subsequently added to the freeze-dried reaction pellets (TwistAmp basic kit, TwistDx, Cambridge, UK), followed by gentle mixing. To initiate the reaction, 2.5 µL of magnesium acetate (MgOAc) was dispensed into the tube's cap and spun down. The fluorescence signal of the RT-RPA reaction was monitored at 1 min intervals using a CFX Opus 96 RT-PCR system (Bio-Rad, CA, USA).

## ELISA assay for antigen and antibody detection

96-well plates were used for ELISA assay. Capture probes, consisting of 1 µg ml$^{-1}$ SARS-CoV-2 nucleocapsid polyclonal antibody (Invitrogen, PA1-41386) for antigen detection and 0.5 µg ml$^{-1}$ antigen S1 (SinoBiological, 40591-V08H) for antibody detection, were prepared in a 10 mM PBS buffer (pH 7.4). A volume of 100 µl of capture probes was added to ELISA plates (BioLegend, 423501) and incubated overnight at 4 °C. Subsequently, the plates were washed three times with 200 µl PBST, followed by the addition of 200 µl of blocking buffer (5% non-fat dry milk) for 1 h. 100 µl of clinical samples (e.g., NPS and serum), diluted in 2.5% non-fat dry milk, were added to well and incubated for 1 h. Detection antibodies, including 1 µg ml$^{-1}$ biotinylated Anti-SARS-CoV-2 nucleocapsid protein antibody (Abcam, ab284656) for antigen detection and including 1 µg ml$^{-1}$ biotin-SP AffiniPure F(ab')$_2$ Fragment Goat Anti-Human IgG (Jackson ImmunoResearch, 109-066-170) for antibody detection, were prepared. A volume of 100 µl of detection antibodies was added to the plates and incubated for 1 h. The plates were then supplemented with 100 µl of streptavidin-HRP (diluted 1:200 in 2.5% blocking buffer), followed by a washing step. Turbo TMB substrate (100 µl; ThermoFisher, 34022) was added and incubated for 20 min, after which 100 µl of 0.1 M H$_2$SO$_4$ in water was added to stop the reaction. The absorbance of the plates was measured at 450 nm using a microplate reader (Safire, Tecan).

## CRISPR-based electrochemical detection of ORF1a gene

Nucleic acids used in this study were synthesized from Integrated DNA Technologies, Inc. (Coralville, IA), Bioneer Co. (Daejeon, Korea), and Panagene (Daejeon, Korea). SARS-CoV-2 virus strain (BetaCoV/Korea/KCDC03/2020) and genomicRNA were obtained from the Korea Disease Control and Prevention Agency (KDCA) and stored at −80 °C until further use. The oligonucleotides for the CRISPR assay were in the following sequence. PNA capture probe: 5'amine-ACAACAACAA-CAACA-3', Reporter probe: 5'-Biotin-AT TAT TAT TAT TAT TTG TTG TTG TTG TTG T-3'. Poly streptavidin-HRP (Thermo Fisher, N200) binds to biotin within reporter probe, leading to localized TMB precipitation. Following RT-RPA amplification without viral RNA extraction, 5 µl of RT-RPA products were added to mixtures containing 42.5 µl CRISPR mix (100 nM Cas12a, 100 nM gRNA) and 50 nM reporter probes. The mixture was incubated for 20 min at 37 °C to activate Cas12a, resulting in cleavage of the reporter probe and preventing its binding to HRP. The chips, which were spotted with PNA capture probes, were incubated with a 40 µl mixture for 5 min, followed by incubation with poly streptavidin-HRP and TMB for 5 min and 2.5 min, respectively. Cyclic voltammetry measurements were performed using a potentiostat (ZIVE SP1, WonATech, Co., Ltd) with a scan rate of 1 V s$^{-1}$.

## Electrochemical detection of SARS-CoV-2 antigen and host antibody

The chips were spotted with capture probes of 1 mg ml$^{-1}$ SARS-CoV-2 nucleocapsid polyclonal antibody (Thermo Fisher, PA1-41386) for antigen detection and 0.5 mg ml$^{-1}$ antigen S1 (SinoBiological, 40591-V08H) for antibody detection. To obtain the calibration curve (Fig. 4b–d), SARS-CoV-2 nucleocapsid protein (GenScript, Z03480) and IgG (Invitrogen, MA5-35939) were selected as target antigen and antibody, respectively. After conjugation, the chips were washed with PBS and quenched using 1 M ethanolamine in PBS. To block any remaining binding sites, a solution of 5% non-fat dry milk in PBS containing 0.05% Tween 20 was applied. Each chip was then used to detect the SARS-CoV-2 nucleocapsid protein and IgG antibody against the immobilized capture probes. For the clinical studies, a total of 60 NPS samples and 53 serum samples were obtained from Gyeongsang National University College of Medicine. All samples were stored at −70 °C until they were used. Informed consent was obtained from all participants for research use. The protocol was reviewed and approved by the Institutional Review Board of Gyeongsang National University College of Medicine in Changwon, Korea (IRB approval number: 2022-10-012). For antigen sensing, 5 µl of clinical NPS was mixed with 20 µl of a 2.5% non-fat dry milk solution. Similarly, for antibody sensing, 2.5 µl of clinical serum was mixed with 22.5 µl of a 2.5% non-fat dry milk solution. These mixtures were then incubated on the chips for 30 min. After rinsing with PBS, 25 µg ml$^{-1}$ detection antibodies (biotinylated Anti-SARS-CoV-2 nucleocapsid protein antibody for antigen detection and biotin-SP AffiniPure F(ab′)$_2$ Fragment Goat Anti-Human IgG for antibody detection) were added to the chips, followed by the addition of diluted poly streptavidin-HRP (1:1000) in 0.1% BSA in PBST for 5 min and TMB for 2.5 min. CV measurements were performed with a scan rate of 1 V s$^{-1}$ to extract the peak oxidation current.

## Multiplexed electrochemical detection of nucleic acid, antigen, and antibody

To detect multiple targets on a chip, four working electrodes (WE) were utilized, each spotted with different capture probes: WE1 with PNA, WE2 with SARS-CoV-2 nucleocapsid polyclonal antibody, WE3 with antigen S1, and WE4 as a BSA control. To determine the optimal dilution ratio of serum spiked into negative NPS (RT-qPCR negative), the spiked NPS was prepared by adding 2 µl of serum to 18 µl of NPS. The prepared spiked NPS was then added to the chip and incubated for 30 min. After rinsing, a solution containing the respective detection antibodies for antigen and antibody sensing was added and allowed to incubate for another 30 min. Simultaneously, NPS without RNA extraction was amplified using RT-RPA for 15 min, as mentioned previously. Following the amplification, 5 µl of the RT-RPA products were added to the CRISPR mix containing reporter probes and incubated on the chip for 20 min at 37 °C. The chips were then incubated with poly streptavidin-HRP and TMB for 5 and 2.5 min, respectively. CV measurements were performed with a scan rate of 1 V s$^{-1}$, and the peak current was calculated using IVMAN 1.5 software.

## Reporting summary

Further information on research design is available in the Nature Portfolio Reporting Summary linked to this article.

## Data availability

All data needed to evaluate the findings can be found in the paper and its Supplementary information. Source data for the figures and Supplementary information are provided as Source data with this paper. Source data are provided with this paper.

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

## Acknowledgements

This work was supported by the funding from the Wyss Institute at Harvard University, KETEP (no. 20183010014470; S.P.) and KEIT (RS-2022-00154853;T.K.) grants funded by the Korea Government (MOTIE), NRF grants funded by Korea Government (MSIT) (NRF-2021M3H4A1A03049049; S.P., NRF-2022R1A2C2006076; S.P., NRF-2021M3E5E3080379; T.K., NRF-2021M3H4A1A02051048; T.K., and NRF-2023R1A2C2005185; T.K.), KEITI grant funded by Korea Government (ME) (2021003370003; T.K.), Nanomedical Devices Development Program of National Nano Fab Center, and KRIBB Research Initiative Program (KGM5472322; T.K.).

## Author contributions

J.C.L., S.Y.K., J.S., H.J., M.K., H.K., S.Q.C., P.J., T.K., S.P., and D.E.I. conceived the study. J.C.L. and J.S. designed the sequence of the oligonucleotides. H.J. conducted the RT-qPCR experiments. J.S. and H.J. performed ELISA assay. J.C.L. and S.Y.K. prepared the emulsion and analyzed the ink properties using techniques such as dynamic light scattering, UV–Vis, and zeta potential. H.K. and S.Q.C. conducted numerical simulations for nozzle-assisted printing. J.C.L., S.Y.K., and M.K. carried out the characterization of the nanocomposite such as SEM, EDS, AFM, and MIP and conducted the electrochemical detection of virus. Validation and reproducibility were performed by J.C.L. and S.Y.K. All authors discussed the results and contributed to writing the manuscript.

## Competing interests

J.C.L, P.J., S.P., and D.E.I. are listed as inventors on patents describing this technology. The remaining authors declare no competing interests.

## Additional information

[1]Wyss Institute for Biologically Inspired Engineering, Harvard University, Boston, MA 02215, USA. [2]Department of Materials Science and Engineering, Korea Advanced Institute of Science and Technology (KAIST), Daejeon 34141, Republic of Korea. [3]Bionanotechnology Research Center, Korea Research Institute of Bioscience and Biotechnology (KRIBB), Daejeon 34141, Republic of Korea. [4]Center for Systems Biology, Massachusetts General Hospital Research Institute, Boston, MA 02114, USA. [5]Department of Radiology, Harvard Medical School, Boston, MA 02114, USA. [6]Department of Chemical and Biomolecular Engineering, KAIST, Daejeon 34141, Republic of Korea. [7]Department of Laboratory Medicine, Gyeongsang National University Hospital, Gyeongsang National University College of Medicine, Jinju-si, Gyeongsangnam-do 52727, Republic of Korea. [8]School of Pharmacy, Sungkyunkwan University (SKKU), Suwon-si, Gyeongi-do 16419, Republic of Korea. [9]Vascular Biology Program and Department of Surgery, Boston Children's Hospital and Harvard Medical School, Boston, MA 02115, USA. [10]Harvard John A. Paulson School of Engineering and Applied Sciences, Harvard University, Cambridge, MA 02138, USA. [11]These authors contributed equally: Jeong-Chan Lee, Su Yeong Kim, Jayeon Song. ✉e-mail: kangtaejoon@kribb.re.kr; stevepark@kaist.ac.kr; don.ingber@wyss.harvard.edu

