## [Peer Review File · Nature Communications]

Reviewers' Comments:

Reviewer #1:

Remarks to the Author:

Authors propose here a nozzle printing method for selectively templating an emulsion-based, micrometer thick, porous nanocomposite conductive coating which offers exceptional antifouling and electroconducting properties and greatly enhanced sensitivity electrochemical biosensors. The experiments are carefully designed and comprehensively discussed and include exhaustive control and characterization experiments which confirmed the competitive advantages of this coating with respect to the other disruptive also proposed by these same authors in ref. 16 to be deposited only on the surface of the working electrode, thus greatly facilitating the manufacture and functionalization of electrochemical sensors for multiplexed and multiomics applications and also to significantly improve the electrochemical performance, sensitivity, and antifouling properties of the resulting biosensors.

The improvement brought to the state of the art by the proposed approach is practically demonstrated by performing individual and multiplexed determination, in connection with different bioreceptors and assay formats, of analytes of different molecular level and of very timely and relevant clinical relevance (SARS-CoV-2 RNA, antigen, and host antibody) in clinical samples with high sensitivity and specificity.

The advances demonstrated in this research are considered tremendously relevant to address hurdles currently restricting the POC diagnostic applications of electrochemical sensors for multiplexed and/or multiomics determinations as well as their use in implantable devices and other healthcare monitoring systems.

Moreover, the results presented are discussed in a very smart but easily understandable way, are very well illustrated and perfectly support the relevant conclusions drawn. Therefore, this pioneering work is recommended for publication of in this leading journal after just discussing the reproducibility of the approach, that is, between the responses provided by different biosensors prepared in the same way, and the storage stability (please indicate the most favorable conditions for it) of the biosensors prepared for the detection of SARS-CoV-2 RNA, antigen, and host antibody without immerse in biofluids. How does this storage stability compare with that of biosensors manufactured on thin nanocomposite-coated electrodes? This is a very important point for the potential commercialization of these biodevices.

Reviewer #2:

Remarks to the Author:

The submitted paper is a follow-on study of the Ingber group's Nature Nanotechnology paper from 2019 on antifouling coatings in electrodes which was quite an impressive body of work. In the method bovine serum albumin with gold nanorods, or graphene, was used to make electrodes that could operate in complex biological fluids. This new paper is essentially reporting on a way of applying these surfaces to electrodes and hence is very much an engineering paper rather than any new conceptual idea. In fact, this referee struggles to see any significant conceptual innovation in the paper. The purported advance is that they can print thicker coatings, using an emulsion approach, and hence get higher signals. The suggested idea is that a thick coating suppresses electrochemistry whilst their printed coatings, which are much more porous, give much higher signals. It sounds quite impressive until one thinks that a porous electrode with a higher surface area gives much higher signal than a flat surface. As such all the paper is really saying is with a higher surface area you get more signal. No surprise there! Sure the thicker, nonporous coatings, give lower electrochemistry than the thinner coatings because the electron transfer pathways through the BSA-gold coatings, require electrons to hop from rod to rod, and the more hopping steps the lower the signal. With the porous coatings, spaces in the layer close to the electrode are accessible so a large proportion of the electrochemical signal could arise from binding events near the underlying electrode where the electron transfer pathlengths are short. The paper is written as though there are many discoveries but most of these are well known. For example, improving the homogeneity in oil droplet size in the emulsions improves the stability is just Ostwald ripening, knowledge that dates back to 1896. The sensing side of the paper is well performed but does not represent an advance in sensing per se either. So the paper is a good body of work but this referee does not feel it possesses the innovation required for Nature

Communications.

Reviewer #3:

Remarks to the Author:

The manuscript by Lee et al. presents a micrometer-thick, porous nanocomposite coating to enhance electrochemical sensors' sensitivity and antifouling properties. This coating is deposited on the sensing electrode with nozzle-assisted printing using an oil-in-water emulsion. The corresponding sensor achieves improved sensitivity and stable current outputs in complex biological fluids. The overall fabrication process appears to be pretty straightforward for electrochemical sensor implementation. Additionally, the practical application of as-prepared sensors is demonstrated by detecting severe acute respiratory syndrome coronavirus 2 (SARS-CoV-2) nucleic acid, antigen, and host antibody in clinical specimens. Despite these interesting aspects, the manuscript still has some obvious issues that need to be addressed:

1. Even though the gold nanowires are a commercial product, the author may still want to provide detailed information, such as diameter and length. To prepare the emulsion, the nanowire mixture is exposed to tip sonication that may sometimes snap the nanowires. Accordingly, the structural integrity of this nanomaterial should be further verified.
2. The reason for including gold nanowires in the nanocomposite coating should be further clarified.
3. The claimed process advantages of nozzle printing lack strong evidence. (a) How to achieve high-quality coating? For example, I wonder whether the printed nanocomposite is spatially uniform. Additional characterizations and discussions are still required. (b) How about the sample-to-sample variations? This point may need some characterizations and statistical analysis.
4. In Fig 3d-e, the evolution of the peak current is used to describe the electrode stability. The thick emulsion-based nanocomposite maintains a stable current as evidence of antifouling characteristics. Despite a steadily decreased current, the thin nanocomposite-coated electrode still exhibits a decent current comparable to a thick emulsion-based nanocomposite electrode after one month. How to justify the advantage of this thick nanocomposite electrode?
5. The sensors' characteristics should be compared with state-of-the-art literature (sensitivity, LOD, antifouling, etc.) to provide a better understanding of their potential.

RESPONSE TO EDITOR AND REVIEWERS

(Lee et al., NCOMMS-23-45926)

REVIEWER #1:

1. Therefore, this pioneering work is recommended for publication of in this leading journal after just discussing the reproducibility of the approach, that is, between the responses provided by different biosensors prepared in the same way,

We sincerely thank to the reviewer for recognizing the quality and strength of our work. In response to reviewer's comment, we have conducted additional experiments to assess the reproducibility.

· Emulsion-based sensor using different biomarkers: We detected anti-GRP78 (anti-glucose-regulating protein 78) and anti-PERK (anti-protein kinase R-like endoplasmic reticulum kinase) using emulsion-based sensors. These antibodies, predominantly found in serum, serve as crucial indicators for assessing the prognosis and advancement of chronic disease such as cancer, diabetes, and rheumatoid arthritis (*Oncotarget* 2017, 8, 24828–24839; *Nature Biotechnology* 2003, 21, 1307–1313). We followed the same protocols for electrode functionalization and electrochemical detection as outlined in the Methods. Utilizing recombinant GRP78 and PERK proteins as recognition elements (*i.e.*, capture probes), we were able to obtain the clear calibration curves for anti-GRP78 and anti-PERK, with a limit of detection of 0.146 ng mL⁻¹ and 0.165 ng mL⁻¹, respectively. These results have been incorporated into the revised text and are presented in Supplementary Fig. 27.

· Nozzle printing reproducibility: We patterned nanocomposites on 10 chips to investigate electrode-to-electrode and chip-to-chip variations in their electrochemical properties and calculated the coefficient of variation (CV) for anodic peak current in potassium ferro/ferricyanide solution. The CV for chip-to-chip and electrode-to-electrode variations were 5.37% and 5.3%, respectively, signifying the robust reproducibility of our coating. We now explain this in the text and added the Supplementary Fig. 7.

2. ...and the storage stability (please indicate the most favorable conditions for it) of the biosensors prepared for the detection of SARS-CoV-2 RNA, antigen, and host antibody without immerse in biofluids. How does this storage stability compare with that of biosensors manufactured on thin nanocomposite-coated electrodes? This is a very important point for the potential commercialization of these biodevices.

In response to the comment, we have investigated the storage conditions of commercial self-diagnostic test kits, including BinaxNow (Abbott), LUCIRA (Pfizer), Ellume COVID-19 home testing (Ellume), and SARS-CoV-2 rapid antibody testing (Rochu). We found that these kits are advised to be stored at temperatures ranging from 2-30 °C with proper packaging. Based on these references, we have conducted a shelf-life test of

emulsion-based sensor and thin nanocomposite sensor (control) in a dry state and N₂ atmosphere at 4 °C and 25 °C. Over a storage period of 7 days, the emulsion-based sensor maintained a peak current at approximately 20 μA, with retention values of 95.34 % at 4 °C and 98.75 % at 25 °C. These values were similar to those observed when stored in PBS (4 °C: 98.75 % and 25 °C: 97.71 %). The thin nanocomposite sensor exhibited relatively consistent electrochemical performance (retention value: 107.70 %) at 4 °C but showed vulnerability at 25 °C with a retention value of 68.01 %. To evaluate the optimized storage stability, we also tested how well the emulsion-based sensor stored in a dry state at 25 °C for 7 days could maintain its sensing performance to detect the IgG antibodies. The emulsion-based sensor demonstrated high sensitivity in detecting 100 pg mL⁻¹ IgG, with a peak current comparable to that of the fresh sensor, meaning that there was no degradation in sensing performance. Hence, our sensor exhibited high stability in a dry state at 25 °C and we have added this information in the text and in the Supplementary Fig. 8.

REVIEWER #2:

1. The submitted paper is a follow-on study of the Ingber group's Nature Nanotechnology paper from 2019 on antifouling coatings in electrodes which was quite an impressive body of work. ... It sounds quite impressive until one thinks that a porous electrode with a higher surface area gives much higher signal than a flat surface. As such all the paper is really saying is with a higher surface area you get more signal. No surprise there!... So the paper is a good body of work but this referee does not feel it possesses the innovation required for Nature Communications.

As the reviewer correctly pointed out, our current paper builds upon this earlier work by presenting an emulsion-based approach to develop porous and micrometer-thick antifouling layer. We also agree with your characterization of our work, as engineering-focused and with some results grounded in established theories like Ostwald ripening. Nevertheless, we still believe that the significance of our contribution in this work lies in an innovative strategy to create the exceptionally durable and uniform patterned-antifouling nanocomposites, which addresses the formidable challenge of reliability issues in electrical signal detection across various biomedical devices. More importantly, the potential clinical and commercial impact of this advance can be huge given the increased stability, robustness, and sensitivity of this coating even relative to our past work. To respond the Reviewer's comment and convey the significance of our technology more clearly, we would like to highlight three key technological advancements achieved through our emulsion-based approach that go beyond the effects of surface area alone.

- **Outstanding stability against non-specific binding:** The advantages arising from the porous nature of the composite not only substantially increase the surface area but also enable one to increase the thickness of the antifouling layer to a microscale. This is a key factor in avoiding non-specific binding, even under long-term exposure to complex

biofluids. Parameters such as density, thickness, and architecture should be considered when enhancing antifouling characteristics. However, it's a double-edge sword, as the maximizing the antifouling effects may inadvertently result in sacrifice of essential functionalities, notably electrode conductivity. In this study, we simultaneously explored the emulsion characteristics and BSA cross-linking, culminating in an ideal structure that maintains electrical signals uniformly in complex biofluids with minimizing the conductivity loss.

To quantify the enhancement of antifouling activities, we exposed both emulsion-based sensors and our thin nanocomposite sensors (used in previous studies cited by the Reviewer) to nasal swab and serum for one month, observing how the signals changed over time (Supplementary Fig. 18a). The emulsion sensor exhibited a coefficient of variation (CV) of 2.3% for nasal swab and 3.1% for serum, while the thin nanocomposite sensor showed higher signal drift with CV of 15.4% and 19.6%, respectively. Considering that the working electrode used in this study has the surface area nine times larger than the electrode used in the previous paper, it could potentially render it more susceptible to fouling. Nevertheless, the emulsion-based sensor demonstrated even more robust antifouling characteristics. We also observed the durability of the two sensors under mechanical stress. Both sensors were immersed in PBS and exposed to an orbital shaker at 150 rpm and the bath sonication for one hour, respectively, with subsequent characterization of changes in electrochemical signals. As shown in the Supplementary Fig. 18b, c, the emulsion-based sensor maintained a similar peak current under both conditions, while the thin nanocomposite sensor exhibited relative vulnerability to external forces ($P = 4.67 \times 10^{-2}$ for shaking and $P = 2.32 \times 10^{-3}$ for sonication).

These advances are of paramount importance in the realm of biosensing as signal drift implies continuous baseline fluctuations. Minimizing such signal drift becomes especially pivotal in decentralized point-of-care diagnostics, where sample processing is limited, allowing for accurate diagnostics irrespective of the duration of specimen exposure. Furthermore, emulsion-based antifouling coatings hold significant value for long-term applications such as implantable devices and stretchable electronics, resolving the electrode malfunction and potentially mitigating inflammatory responses arising from non-specific adsorption. We have edited the text and added the signal drift analysis as Supplementary Fig. 18.

· **Continuous nozzle printing resulting from emulsion development:** In the previous study, drop-casting was utilized to coat a BSA-based aqueous solution. While drop-casting is simple and suitable for creating uniform films at the 10 nm level, it lacks continuous printing capabilities and exhibits decreased uniformity when increasing thickness. Despite our efforts to apply nozzle printing for precise control of composites, as depicted in the Supplementary Fig. 4, BSA-based aqueous solution showed Newtonian characteristics, posing challenges in achieving uniform printing due to rapid

solution extrusion and drop splitting. In contrast, our optimized emulsion exhibits shear thinning, a non-Newtonian property, owing to the presence of homogeneous oil droplets. This enables moderate solution extrusion by inducing viscosity changes within the narrow nozzle. Therefore, emulsion development enabled the application of continuous printing that was previously unattainable, further facilitating high-resolution patterning on the working electrode, which has great potential commercial value. We have addressed this point more clearly in the text, revised Supplementary Fig. 4, and added Supplementary Fig. 1.

· **Accurate and simultaneous detection of SARS-CoV-2 nucleic acid, antigen, and host antibody**: Leveraging the advantages of the highly stable and sensitive emulsion-based sensor in complex specimens, we conducted the first comprehensive analysis of nucleic acid, antigen, and serological diagnoses in clinical samples. As shown in Fig. 4k, observing the changes in the three targets in each patient and comparing their trends can significantly enhance testing accessibility and play a crucial role in formulating treatment strategies.

In response to the Reviewer's concern, we now more clearly describe these and other features that clearly demonstrate novelty of our work in the Introduction and Discussion.

REVIEWER #3:

1. Even though the gold nanowires are a commercial product, the author may still want to provide detailed information, such as diameter and length. To prepare the emulsion, the nanowire mixture is exposed to tip sonication that may sometimes snap the nanowires. Accordingly, the structural integrity of this nanomaterial should be further verified.

We now include more detailed information regarding the gold nanowires in the Methods. In response to your concerns, we have conducted additional experiments to characterize the gold nanowire structure before and after tip sonication. As seen in the Supplementary Fig. 9, the gold nanowires maintained a nearly consistent diameter and the length after tip sonication without any breakage. Previous studies have indicated that noticeable fragmentation of the metal nanowires typically occurs after 1 or 2 hours of sonication (*Micromachines* 2019, 10, 29; *ACS Nano* 2020, 14, 11, 15286–15292). Additionally, since the force induced by ultrasonic waves increases in relation to the length of nanowire, relatively long nanowires ($L > 20 \mu\text{m}$) are more prone to bending and subsequent breakage (*RSC Adv.*, 2016, 6, 72080; *ACS Nano* 2020, 14, 11, 15286–15292). Therefore, we concluded that there was no significant deformation in the nanowires, as our optimized sonication time (*i.e.*, 25 minutes) is insufficient to induce breakage and the nanowire length is relatively short (*i.e.*, $4.5 \mu\text{m}$). We now clarify these points and have added these results as Supplementary Fig. 9. We also verified the structural integrity of the gold nanowires by

analyzing the EDS in the Fig. 2, as well as carrying out SEM and the TOF-SIMS spectrum analysis as shown in the revised Supplementary Fig.12.

2. The reason for including gold nanowires in the nanocomposite coating should be further clarified.

Conductive nanomaterials play a vital role in enhancing overall electrical performance within antifouling coatings because most antifouling materials are insulating. Therefore, incorporating conductive nanomaterials like AuNW and rGO is required to effectively enhance electron transfer to the underlying electrode. We now explain this more clearly in the Introduction.

3. The claimed process advantages of nozzle printing lack strong evidence. (a) How to achieve high-quality coating? For example, I wonder whether the printed nanocomposite is spatially uniform. Additional characterizations and discussions are still required.

As shown in the Supplementary Fig. 4, the emulsion exhibits shear-thinning behavior, where viscosity decreases with increasing shear rate, in comparison to the control solution. This property is vital for stable ejection of ink through the narrow nozzle, allowing for high-quality printing without particle aggregation, a result corroborated by computational fluid dynamics (CFD) simulations. We conducted additional analysis of the spatial uniformity of the coating using confocal microscopy to quantify fluorescence intensity of FITC-labeled anti-IgG that we immobilized on the nanocomposite surface. The coefficient of variation (CV) of fluorescence intensity within the printed composite was 7.81 %, demonstrating high uniformity of the density of capture probe along the surface of the nanocomposite. We also assessed the patterning uniformity of the nanocomposite by comparing the center-to-center distance of the patterned composites. We edited the text and added the Supplementary Fig. 5 to explain this more clearly.

4. (b) How about the sample-to-sample variations? This point may need some characterizations and statistical analysis.

To address the reviewer's comments, we patterned nanocomposites on 10 electrochemical chips and compared their electrochemical characteristics to investigate the sample-to-sample variations. As a result, the electrode-to-electrode and the chip-to-chip variations were 5.3% and 5.37%, respectively, indicating minimal changes in electrochemical properties attributed to the high printing uniformity. We have edited the text and added the Supplementary Fig. 7.

5. In Fig 3d-e, the evolution of the peak current is used to describe the electrode stability. The thick emulsion-based nanocomposite maintains a stable current as evidence of antifouling characteristics. Despite a steadily decreased current, the thin nanocomposite-coated electrode still exhibits a decent current comparable to a thick emulsion-based nanocomposite electrode after one month. How to justify the

advantage of this thick nanocomposite electrode?

Despite the thin nanocomposite sensor exhibiting significant current in a month-long exposure to complex biofluids, its signal undergoes continuous changes over time. This continuous baseline fluctuation makes it challenging to distinguish target-specific signal from non-specific signals. On the other hand, emulsion-based sensors, with their ability to sustain a consistent peak current, minimizing signal drift and thus enabling accurate detection of target analytes in complex specimens. To clearly convey the peak current changes in complex biological fluids, we analyzed the retention values of peak currents in 1% BSA, NPS, and serum for both thin nanocomposite sensor and emulsion-based sensor. We now clarify this point in the text and show the results in Supplementary Fig.17.

6. The sensors' characteristics should be compared with state-of-the-art literature (sensitivity, LOD, antifouling, etc.) to provide a better understanding of their potential.

To provide a better understanding between our work and state-of-the-art literature, we have updated Table S6 comparing detection methods, target molecules, sensitivity, and antifouling activities for each technique.

Reviewers' Comments:

Reviewer #1:

Remarks to the Author:

The authors have thoroughly considered all my previous comments. Therefore, I am pleased to recommend the publication of this revised manuscript in this leading journal.

Reviewer #3:

Remarks to the Author:

Most issues have been adequately addressed through the revision. However, I still have a minor suggestion regarding Supplementary Figure 9. The authors may want to supplement the results with low-magnification images to provide a more comprehensive view of the nanowires. This would help the readers to verify that the length has not been altered through sonication. Except for that, the rest of the manuscript may be suitable for further consideration towards publication.

RESPONSE TO EDITOR AND REVIEWERS

(Lee et al., NCOMMS-23-45926)

REVIEWER #3:

1. However, I still have a minor suggestion regarding Supplementary Figure 9. The authors may want to supplement the results with low-magnification images to provide a more comprehensive view of the nanowires. This would help the readers to verify that the length has not been altered through sonication.

In response to your suggestion, we have revised the Supplementary Figure 9 by incorporating SEM images with low-magnification to offer a comprehensive view of the nanowires. The nanowires maintained a nearly consistent diameter and the length after tip sonication without breakage.